# Xenon iron oxides predicted as potential Xe hosts in Earth's lower mantle

Feng Peng [1,2,8], Xianqi Song [3,4,8], Chang Liu[4,5,6], Quan Li [3,4,5,6✉], Maosheng Miao [2], Changfeng Chen [7✉] & Yanming Ma [3,4,5✉]

An enduring geological mystery concerns the missing xenon problem, referring to the abnormally low concentration of xenon compared to other noble gases in Earth's atmosphere. Identifying mantle minerals that can capture and stabilize xenon has been a great challenge in materials physics and xenon chemistry. Here, using an advanced crystal structure search algorithm in conjunction with first-principles calculations we find reactions of xenon with recently discovered iron peroxide $FeO_2$, forming robust xenon-iron oxides $Xe_2FeO_2$ and $XeFe_3O_6$ with significant Xe-O bonding in a wide range of pressure-temperature conditions corresponding to vast regions in Earth's lower mantle. Calculated mass density and sound velocities validate Xe-Fe oxides as viable lower-mantle constituents. Meanwhile, Fe oxides do not react with Kr, Ar and Ne. It means that if Xe exists in the lower mantle at the same pressures as $FeO_2$, xenon-iron oxides are predicted as potential Xe hosts in Earth's lower mantle and could provide the repository for the atmosphere's missing Xe. These findings establish robust materials basis, formation mechanism, and geological viability of these Xe-Fe oxides, which advance fundamental knowledge for understanding xenon chemistry and physics mechanisms for the possible deep-Earth Xe reservoir.

[1] College of Physics and Electronic Information & Henan Key Laboratory of Electromagnetic Transformation and Detection, Luoyang Normal University, 471022 Luoyang, China. [2] Department of Chemistry and Biochemistry, California State University Northridge, Northridge, CA 91330-8262, USA. [3] State Key Laboratory of Superhard Materials, College of Physics, Jilin University, 130012 Changchun, China. [4] Innovation Center for Computational Methods & Software, College of Physics, Jilin University, 130012 Changchun, China. [5] International Center of Future Science, Jilin University, 130012 Changchun, China. [6] Key Laboratory of Automobile Materials of MOE and Department of Materials Science, College of Materials Science and Engineering, Jilin University, 130012 Changchun, China. [7] Department of Physics and Astronomy, University of Nevada, Las Vegas, NV 89154, USA. [8] These authors contributed equally: Feng Peng, Xianqi Song. ✉email: liquan777@jlu.edu.cn; chen@physics.unlv.edu; mym@jlu.edu.cn

The chemical reaction of inert xenon (Xe), a quintessential full-shell element, was earliest predicted by Pauling in 1933 and the first xenon compound was experimentally produced in 1962[1]. Then, more xenon compounds were experimentally synthesized at ambient pressure, containing some most electronegative atoms like fluorine[2–5] or oxygen[6–9]. Subsequently, scientists found that pressure can effectively improve the chemical reactivity of Xe[10–17]. At moderate pressures, solid xenon forms weakly bonded compounds with other species, e.g., with $H_2O$[10] and $O_2$[11,12] at 1 and 3 GPa, respectively. Strikingly, several novel Xe compounds with unusual stoichiometries are found to be thermodynamically stable at high pressures, e.g., Xe oxides[13,14], Xe nitrides[15], xenon-hydrogen[16], and Xe–Mg compounds[17].

At ultra-high-pressure conditions, the high volatility, relative chemical inertia, and abundant isotopes of xenon make it a valuable tracer in the study of evolutionary dynamics and history of Earth. However, 99% of Earth's primordial Xe is mysteriously missing as characterized by its very low abundance compared to other noble gases in Earth's atmosphere[18], which is known as the missing Xe problem[19]. Early hypotheses proposed that Xe might have escaped from the atmosphere after ionization[20–22], or that it might be stored in the interior of Earth[23–29]. Attempts to incorporate Xe into ices, clathrates and sediments in the Earth's crust were not successful[27–29]. Laboratory experiments have succeeded in trapping Xe in quartz[30,31] and observing predicted stable xenon oxides[13,14,32]; but these results cannot explain the missing Xe mystery, because xenon oxides are unstable in equilibrium with metallic iron in lower mantle while xenon silicates decompose spontaneously at mantle pressures[13]. Reactions of Xe with iron and nickel in Earth's core were predicted as a viable scenario[33] and the predicted compounds were synthesized under core pressure and temperature conditions[32,34]. However, it remains highly intriguing and challenging to explore possible capture and stabilization of Xe by suitable minerals in Earth's mantle, which is of special significance because it was estimated that the loss of atmospheric Xe occurred about 100 million years from accretion, corresponding to the time of mantle differentiation event[30].

Extensive past searches were unable to find chemical reactions of Xe with known mantle minerals. Recently discovered $FeO_2$ synthesized at lower mantle conditions[35] and stablized above 74 GPa in theoretical calculation[36], offer an intriguing new possibility. This newly identified iron peroxide is able to react with helium to form a rare helium-bearing compound that explains deep-Earth primordial helium deposits revealed by geochemical evidences[37]. This finding raises exciting prospects that $FeO_2$ may be able to react with Xe (actually $P–T$ stability range of $FeO_2$ has not been completely established in experiments) at mantle conditions, thereby forming compounds capable of trapping Xe in Earth's interior. In this work, we have explored possible reactions of Xe with $FeO_2$ in contrast with known mantle constituents FeO, $SiO_2$, MgO, CaO, and $Al_2O_3$. We find that $FeO_2$ has unique ability to react with Xe and form robust Xe-Fe oxides $Xe_2FeO_2$ and $XeFe_3O_6$ with surprisingly strong Xe–O bonding, while other mantle oxides do not react with Xe. We have further examined mass density and sound velocities of these Xe-Fe oxides, and the results support their viability in vast lower mantle region. These findings establish robust materials basis, formation mechanism, and geological viability of these Xe-Fe oxides, which advance fundamental knowledge for understanding xenon chemistry and physics mechanisms for the possible deep-Earth Xe reservoir.

## Results

**Crystal structures.** We take the crystal phases identified by the structure search process at various $FeO_2$: Xe ratios and compute their enthalpy to determine the most viable structure at each composition, and based on the obtained results we construct the convex hull, as shown in Fig. 1a, which indicates stable products from reactions of $FeO_2$ and Xe. This exercise has led to the discovery of two Xe-Fe oxides, $Xe_2FeO_2$ and $XeFe_3O_6$, that are stable against decomposition at 150 GPa and 200 GPa. The pressure–volume terms, associated with packing efficiency, make the major contribution to guarantee the thermodynamical stability of $Xe_2FeO_2$ and $XeFe_3O_6$ with formation enthalpies lying on the convex hull (Supplementary Fig. 1). These two Xe-Fe oxides are both still stable relative to all possible binary phases or pure simple substances of $Xe-Fe-O_2$, which can be seen in Supplementary Information (Supplementary Figs. 2 and 3). The details Phonon dispersions calculated at 150 GPa (Fig. 1b, c) show that these compounds are dynamically stable and, as will be shown below, thermal effects further stabilize both oxides over a wider range of pressure at elevated temperatures at deep-Earth conditions. Here we first present a full analysis of structural and bonding characters of these two oxides at 150 GPa as a representative case study. The compound $Xe_2FeO_2$ is crystalized in a monoclinic structure with $P2_1/c$ symmetry (Fig. 1d); its structural motif consists of stacked layers of corner-sharing octahedron with each Fe atom surrounded by six O atoms and the Fe atom is centered in a slightly distorted octahedron containing Fe–O bond lengths in a narrow range of 1.79–1.82 Å at 150 GPa. Each Xe atom in this structure has a coordination number of 3, bonding at the corners of $FeO_6$ octahedra with the Xe–O bond lengths in the range of 2.40–2.42 Å at 150 GPa, which are similar to those found in $Xe_2O_3$ (~2.50 Å) and $Xe_2O_5$ (~2.37 Å) at the same pressure[38]. Meanwhile, $XeFe_3O_6$ is stabilized in a triclinic structure with $P\text{-}1$ symmetry, containing two formula units per cell (Fig. 1e); its corner-sharing $FeO_6$ octahedra host Fe–O bonds with lengths of 1.73–1.81 Å at 150 GPa, forming a tubular structure, and each Xe atom has a coordination number of 6, located in the Fe-O tube with the nearest Xe-O distance of 2.08 Å, resulting in the vibrational mode of the lowest-frequency branch at F as shown in Fig. 1c. Further vibrational analyses are shown in Supplementary Fig. 4 and structural details of both Xe-Fe oxides at 150 GPa are listed in Supplementary Information (Supplementary Table 1).

## Discussion

**Chemical Bonding.** To assess bonding characters in the two Xe-Fe oxides, we have calculated their electronic density of states (DOS) at 150 GPa. The results shown in Fig. 2a, c reveal metallic nature of these compounds; crucially, in both cases the DOS in the vicinity of the Fermi level contain significant contributions from the Fe 3d, Xe 5p and O 2p states, indicating considerable Fe–O and Xe–O bonding interactions. We further calculated projected crystal orbital Hamiltonian population (pCOHP) that evaluates weighted population of wavefunctions on two atomic orbitals of a pair of selected atoms[39]. The results in Fig. 2b, d reveal characteristic Fe–O and Xe–O covalent bonding as indicated by the prominent features of strong low-energy bonding states together with some occupied antibonding states near the Fermi level in each case. It is noted that the occupied bonding states in $Xe_2FeO_2$ occur deeper below the Fermi level compared to those in $XeFe_3O_6$, indicating higher stability of $Xe_2FeO_2$. Moreover, integrated COHP (ICOHP) provides an estimate of the overall bonding strength[39]. Calculated ICOHP values for the Fe–O and Xe–O bonds at 150 GPa are -1.45 eV/pair and −0.24 eV/pair in $XeFe_3O_6$ and −1.01 eV/pair and −0.12 eV/pair in $Xe_2FeO_2$, respectively. These results show considerable Xe–O bonding compared to the strong Fe–O bonding, in sharp contrast to recently discovered He-$FeO_2$ compound where He atoms show little direct bonding[37] but instead serve as a Coulomb shield in

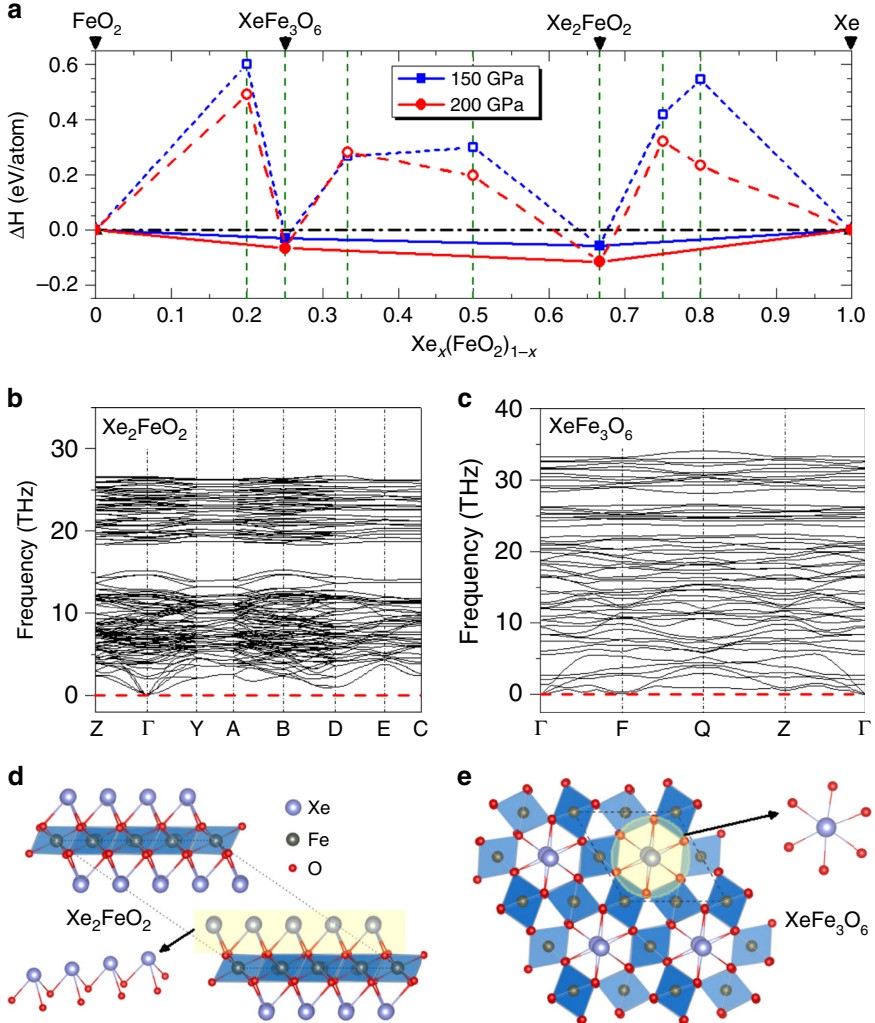

**Fig. 1 Energetic stability and structures of Xe-Fe oxides. a** The ground-state convex hull with solid lines for the FeO$_2$-Xe system constructed from calculated formation enthalpy ($\Delta H$) data, identifying two stable Xe-Fe oxides, Xe$_2$FeO$_2$ and XeFe$_3$O$_6$, at 150 GPa and 200 GPa. The solid and open symbols represent the stable structures lying on the convex hull and the unstable structures above the convex hull, respectively. **b** Phonon dispersions of Xe$_2$FeO$_2$ at 150 GPa. **c** Phonon dispersions of Xe$_3$FeO$_6$ at 150 GPa. **d** The structure of Xe$_2$FeO$_2$ with a polyhedral and an enlarged view. **e** The structure of Xe$_3$FeO$_6$ with a polyhedral and an enlarged view. Xe, Fe, and O atoms are, respectively, shown in purple, gray, and red spheres.

stabilizing the structure[40]. We also examined charge distribution in Xe$_2$FeO$_2$ and XeFe$_3$O$_6$ by a Bader charge analysis[41], and the results reveal a considerable amount of highly unusual charge transfer from Fe and Xe to O atoms. At 150 GPa, the Bader partial charges in Xe$_2$FeO$_2$ are +0.30, +1.40, −1.00 for Xe, Fe, and O, respectively; meanwhile, Bader partial charges in XeFe$_3$O$_6$ are +1.35, +1.35, −0.90 for Xe, Fe, and O, respectively, at the same pressure. As a result, Xe atoms in XeFe$_3$O$_6$ can donate more electrons than in Xe$_2$FeO$_2$ and Xe atoms can display different valence states in FeO$_2$–Xe compounds. These significant charge transfers once again indicate strong bonding formation involving Xe, which is rare among noble gases atoms.

**Thermal effect**. Thermal effects play a crucial role in material stability at pertinent geophysical conditions, where temperatures reach 2000–4500 K. Here, we evaluate Gibbs free energy of Xe$_2$FeO$_2$ and XeFe$_3$O$_6$ by calculating lattice contributions to the entropic term using the quasiharmonic approximation to account for volume dependence of phonon frequencies due to temperature induced lattice expansion. In Fig. 3a we present relevant energetic terms affecting structure stability. It is seen that internal energy $U$ values of the two Xe-Fe oxides are higher than those of

their separate constituents, namely Xe and FeO$_2$, but the $PV$ terms are decisively favorable and dominant, producing the lower enthalpy $H$ for the formation of both oxides. The temperature effect (i.e., thermal vibration of atomic positions) are favorable to reduce Gibbs free energy $G$ of the Xe-Fe oxides even more relative to their separate constituents, further stabilizing the resulting crystal structures. Consequently, the threshold pressure above which these oxides are stable reduces considerably at increasing temperatures, thereby significantly expanding their stability field as will be seen in the phase diagram presented below.

**Phase diagram**. For a full assessment of temperature effects, we have performed extensive energetic and ab initio molecular dynamics (AIMD) simulations to evaluate structural stability and construct pressure–temperature ($P$–$T$) phase diagram for the Xe-Fe oxide system. We present in Fig. 3b, c the mean square deviations (MSD) of atomic positions in the Xe-Fe oxides at typical high $P$–$T$ conditions of 150 GPa and 3000 K, and the results show that the Fe, O, and Xe atoms all remain near their lattice sites, indicating stability of the crystal structure. Similar AIMD simulations were performed systematically to probe each phase and determine the boundary where temperature-driven

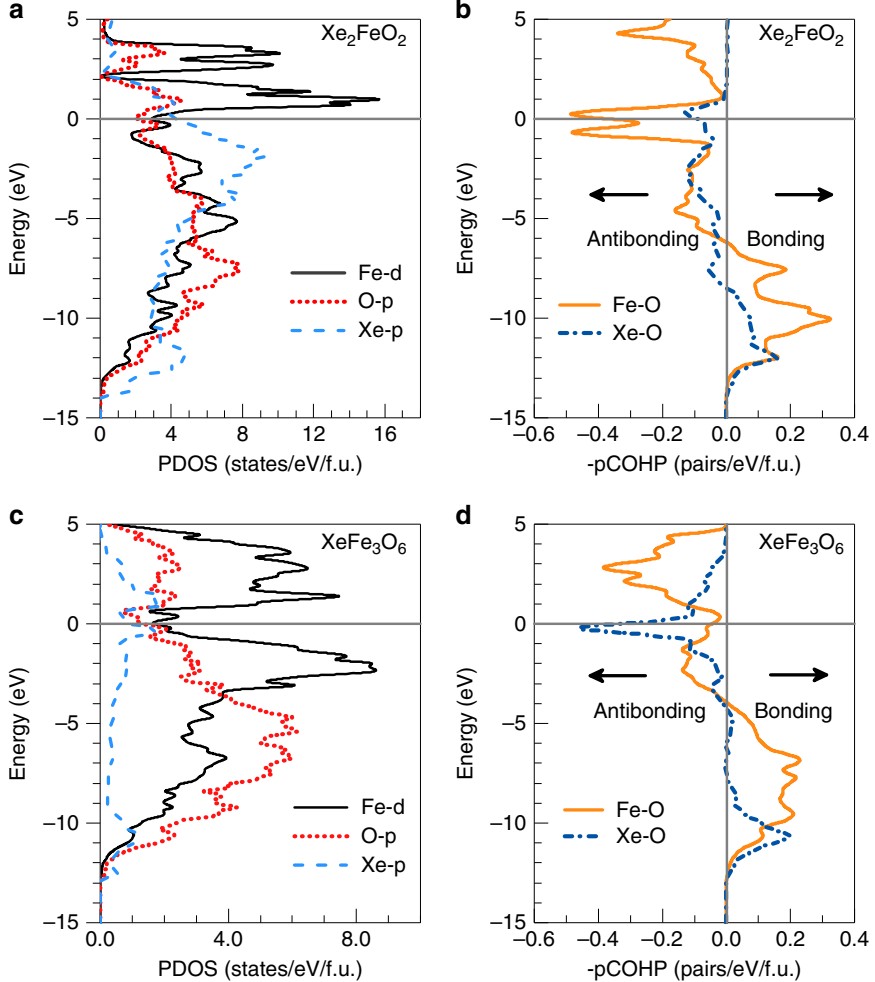

**Fig. 2 Electronic properties of the two Xe-Fe oxides at 150 GPa. a** Projected density of states (PDOS) of Fe-d, O-p, and Xe-p orbitals in $Xe_2FeO_2$.
**b** Projected crystal orbital Hamiltonian Population (-pCOHP) of the newly predicted $Xe_2FeO_2$ compound. The values of -pCOHP >0 signify bonding states and the values of -pCOHP <0 signify antibonding states. **c** PDOS of Fe-d, O-p, and Xe-p orbitals in $XeFe_3O_6$. **d** -pCOHP of the newly predicted $XeFe_3O_6$ compound. The Fermi energy is set to zero of the energy.

instability sets as indicated by deviating MSD from equilibrium positions; meanwhile Gibbs free energies were computed and compared to determine the boundaries between different solid phases in the P–T space. The resulting phase diagram (Fig. 4) spans a wide P–T range covering the lower mantle and higher P–T regions.

We now analyze the stability fields of the predicted Xe-Fe oxides under the (P, T) conditions conforming to geological constraints dictated by the geotherm that is also shown in Fig. 4. It is seen that $Xe_2FeO_2$ is stable in the pressure range 110-120 GPa and temperatures around 2500 K inside the geotherm corresponding to the deep lower mantle region; meanwhile, both $Xe_2FeO_2$ and $XeFe_3O_6$ are stable between pressures 120–136 GPa and temperatures 2500–3600 K inside the geotherm corresponding to the lowest mantle to core-mantle boundary (CMB); finally, as pressure and temperature rise further, $Xe_2FeO_2$ becomes the sole stable phase.

The above results suggest stable Xe-Fe oxides under the (P, T) conditions in vast deep-Earth regions. It is, however, necessary to assess the viability of the predicted Xe-Fe oxides in geological environments by examining their key material characteristics. To this end, we have run AIMD simulations to determine crystal structures at selected (P, T) conditions and used an AIMD-based strain-stress method[42,43] to calculate the elastic tensors, from

which elastic-wave velocities were determined by solving the Christoffel equation det $|T_{ik}\text{-}\delta_{ik}\rho V^2| = 0$, where $\delta_{ik}$ is the Kronecker delta function, V is one of the seismic velocities, and $T_{ik}$ is the Christoffel stiffness[44].

**Mass density and sound velocities**. We examine mass density and mean compressional (P-wave) and shear (S-wave) sound velocities, $V_P$ and $V_S$, respectively, at two representative (P, T) points: (120 GPa, 2500 K) for lower mantle, and (135 GPa, 3500 K) for CMB, and compare with geological data. We first examine $XeFe_3O_6$, whose stability field compared to the geotherm indicates its stability in the lower mantle and CMB regions, but higher temperatures destabilize this compound. The calculated densities of $XeFe_3O_6$ are 8.86 and 9.06 g/cm³ at the selected lower mantle and CMB (P, T) points, respectively, which lie within or close to the range of 4.95–9.90 g/cm³ from the core rigidity zone (CRZ) model and the range of 5.57–8.91 g/cm³ from the ultralow velocity zone (ULVZ) model[45]. The calculated $V_P$ ($V_S$) are 8.94(4.03) km/s and 9.20(4.26) km/s, respectively, which lie within the range of 8.20–10.70 km/s (1.00–5.00 km/s) for the ULVZs[45]. All these results indicate that $XeFe_3O_6$ is a viable constituent at the lower mantle and CMB (P, T) conditions.

The calculated densities of $Xe_2FeO_2$ are 9.78 and 9.87 g/cm³ at the lower mantle and CMB points, respectively, which lie outside

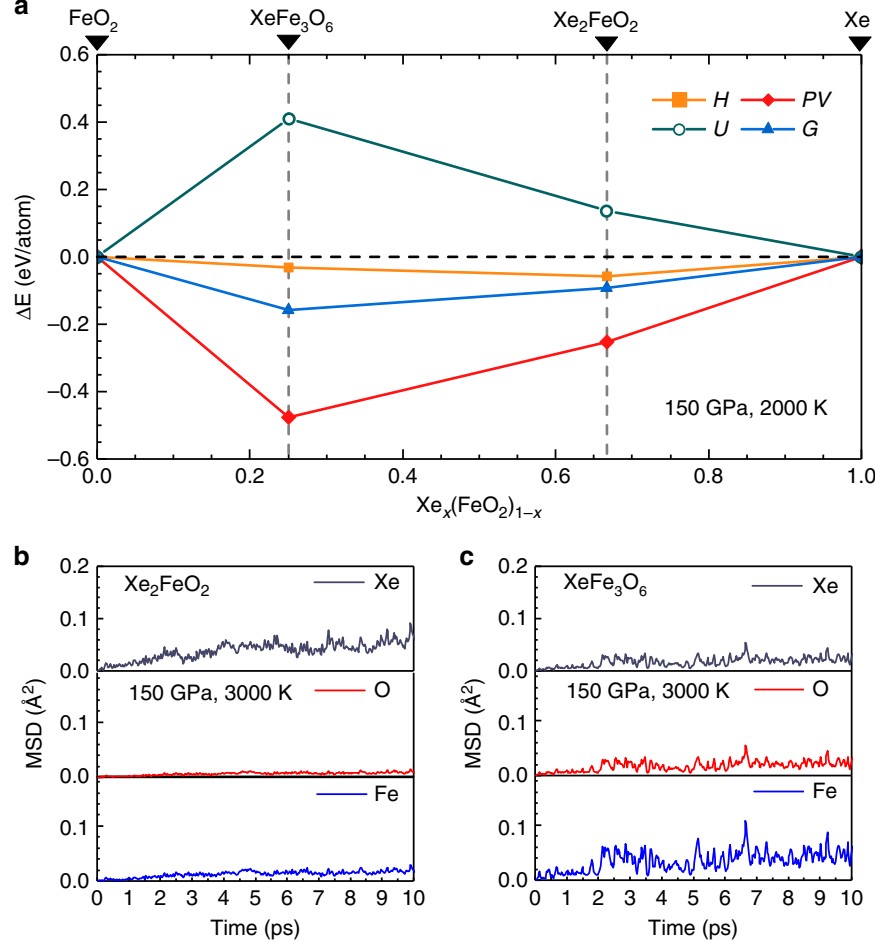

**Fig. 3 Calculated energetic terms and the mean square deviations. a** Various energetic term ($\Delta E$) for the two Xe-Fe oxides, enthalpy ($H$), pressure–volume ($PV$) terms, internal energy ($U$), and Gibbs free energy ($G$) at 150 GPa, 2000 K. **b** Mean square deviations (MSD) of Xe, O, and Fe atomic positions in $Xe_2FeO_2$ at 150 GPa, 3000 K. **c** MSD of Xe, O, and Fe atomic positions in $XeFe_3O_6$ at 150 GPa, 3000 K.

the range of 5.57–8.91 g/cm$^3$ from the ULVZ model and at the top of the range of 4.95–9.90 g/cm$^3$ from the CRZ model[45]. The calculated $V_P$ ($V_S$) are 7.83(4.21) km/s and 7.94(4.02) km/s at the lower mantle and CMB points, respectively. While these $V_S$ values lie within the range of 1.00–5.00 km/s from the CRZ model or the range 2.91–6.17 km/s from the ULVZ model[45], the $V_P$ values are out of the range of 10.97–13.03 km/s from the ULVZ model or the range of 8.20–10.70 km/s from the CRZ model[45]. These results render $Xe_2FeO_2$ a marginal lower mantle or CMB constituent at best.

**Reaction of noble gases and deep-Earth constituents.** Finally, we highlight several significant aspects on the special role of $FeO_2$ in trapping Xe in deep Earth. First, we have systematically examined possible reaction of Xe with major deep-Earth constituents FeO, $SiO_2$, MgO, CaO, and $Al_2O_3$, and the resulting convex-hull data (Supplementary Fig. 5) show highly unfavorable energetics in all the cases, offering an underlying cause for unsuccessful past attempts to find Xe-bearing minerals in Earth's mantle. Second, we have examined possible reactions of other noble gases Ne, Ar, and Kr with $FeO_2$, and the results (Supplementary Fig. 6) indicate no tendency toward forming any stable noble-gas-Fe oxides up to 200 GPa. These results provide the possibility that Xe could be the sole inert element for reacting with deep-Earth constituents under mantle conditions. Moreover, while He-bearing compound $FeO_2He$ is found stable at CMB conditions, there is little direct bonding between He and Fe or O

atoms in the compound[37]. Compared to Kr, Ar, Ne, and He, Xe has the lowest ionization energy and electronegativity, and consequently Xe is the easiest noble-gas atom to open up its outermost closed shell and form direct bonding as found in $Xe_2FeO_2$ and $XeFe_3O_6$.

In summary, we have identified two Xe-Fe oxides, $Xe_2FeO_2$ and $XeFe_3O_6$, that are the first viable Xe-bearing compound at Earth's lower mantle conditions. These new compounds are predicted by extensive crystal structure search in conjunction with ab initio energetic calculations and molecular dynamics simulations. Mass densities and compressional and shear sound velocities calculated at deep-Earth conditions are compatible with pertinent ULVZ and PREM data, thus confirming viability of $Xe_2FeO_2$ in geological environments. These results provide compelling evidence for a distinct deep-Earth Xe reservoir beyond previously proposed Xe-Fe and Xe-Ni intermetallic compounds in Earth's inner core, thereby greatly expanding the range and scope of Xe-bearing compounds in deep Earth. The Xe-Fe oxides may enrich the understanding of prominent geophysical and geochemical processes, such as seismic anomalies near the CMB and possibly new chemical reactions inside Earth's lower mantle.

## Methods

**Structural predictions.** Our structure search is based on a global optimization of free-energy surfaces using the CALYPSO methodology[46,47], which has been successfully employed in predicting a large variety of crystal structures[48–52]. Evolutionary variable-cell calculations were performed at 120, 150, and 200 GPa with 1,

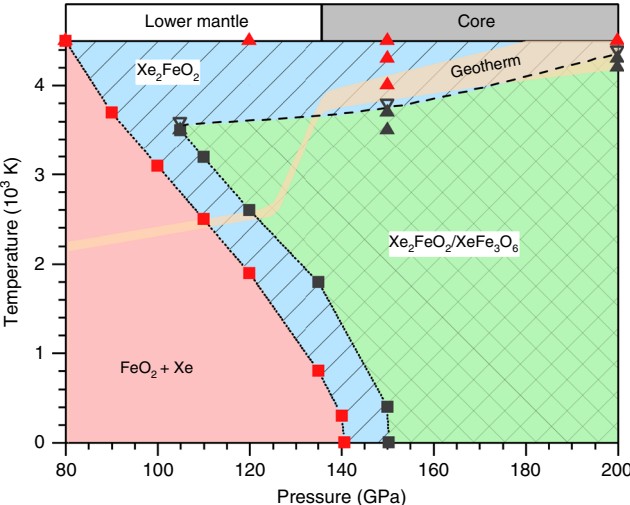

**Fig. 4 The Pressure–temperature (P–T) phase diagram of the Xe-FeO₂ system.** The dotted and dashed lines indicate phase boundaries and temperature-driven phase instability, which are determined by Gibbs free energy and ab initio molecular dynamics (AIMD) calculations, respectively. The square symbols show phase transition points based on relative Gibbs free energy, and the solid and open triangle symbols represent stable and temperature-driven unstable solid phases, respectively, determined by AIMD simulations. The P–T regions of stable $Xe_2FeO_2$ are covered by slash lines and the P–T regions of stable $XeFe_3O_6$ are filled by back-slash. The pressure boundary between the lower mantle and the core is shown at the top of the figure. Also shown is the geotherm of the Earth's interior[65].

2, 3, and 4 formula units (f. u.) per cell. Most searches converge in 50 generations with about 2500 structures generated.

**Ab initio calculations.** First-principles total-energy and electronic property calculations were carried out using the density functional theory with the Perdew–Burke–Ernzerhof exchange-correlation functional in the generalized gradient approximation (GGA)[53,54] as implemented in the VASP code[55], adopting frozen-core all-electron projector-augmented wave method[56] with $3s^2\,3p^6\,3d^7\,4s^1$, $2s^2\,2p^4$, and $4d^{10}\,5s^2\,5p^6$ treated as valence electrons for Fe, O, and Xe, respectively. Correlation effects among the Fe 3d electrons were treated in the GGA + U approach[57,58], adopting the recently proposed on-site Coulomb interaction $U = 5.0$ eV and a Hund's coupling $J = 0.8$ Ev[36,59–61]. The spin-polarized and magnetic states were considered in obtaining the total-energy of the compounds containing iron. Zero-point energy was included in all reported calculations. A cutoff energy of 1200 eV for the plane-wave expansion and fine Monkhorst-Pack **k** meshes[62] were chosen to ensure enthalpy convergence of better than 1 meV/atom.

**Phonon calculations.** To determine the dynamical stability, we performed phonon calculations by the direct supercell method using the Hellmann-Feynman theorem, as implemented in Phonopy code[63]. The harmonic interatomic force constants are calculated by $3 \times 3 \times 3$ and $3 \times 2 \times 2$ supercells for $P2_1/c$-$Xe_2FeO_2$ and $P\bar{1}$-$XeFe_3O_6$, respectively. Forces were calculated for atomic displacements of 0.01 Å, with a convergence threshold of $1 \times 10^{-5}$ eV/Å.

**Van der Waals interaction.** To examine the contribution of vdW interaction to the lattice energy, we have calculated the enthalpy of formation of $Xe_2FeO_2$ at high pressures using the vdW-DF2 density functional[64]. Our results show that the enthalpy of formation is less sensitive to the contribution of vdW correction at high-pressure conditions for Xe-FeO₂ compounds, e.g., about 0.6 meV/atom for $P2_1/c$ $Xe_2FeO_2$ at 135 GPa, thus vdW interaction is not considered in the calculations of lattice energy.

## Data availability

The authors declare that the main data supporting the findings of this study are contained within the paper and its associated Supplementary Information. All other relevant data are available from the corresponding author upon reasonable request.

## Code availability

CALYPSO code is free for academic use, by registering at http://www.calypso.cn. The other scripts are available from the authors upon reasonable request.

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

## Acknowledgements

We acknowledge funding from National Natural Science Foundation of China under Grant Nos. 11774140 and 11622432, China Postdoctoral Science Foundation under Grant No. 2016M590033, Natural Science Foundation of Henan Province under Grant No. 162300410199, Program for Science and Technology Innovation Talents in University of Henan Province under Grant No. 17HASTIT015, and Open Project of the State Key Laboratory of Superhard Materials, Jilin University under Grant No. 201602.

## Author contributions

Q.L., C.C., and Y.M. designed the research; F.P., and X.S. performed the calculations; F.P., X.S., C.L., Q.L., M.M., C.C., and Y.M. analyzed and interpreted the data, and contributed to the writing of the paper.

## Competing interests

The authors declare no competing interests.
