## [Peer Review File · Nature Communications]

REVIEWER COMMENTS

Reviewer #1 (Remarks to the Author):

This difference between expected and actual abundance of xenon in the Earth's atmosphere is known as the "missing xenon paradox", which has left scientists stumped for decades. Attempt to explain of the paradox, many hypotheses have been proposed, e.g., Xe might be located within the Earth's as stable compounds. However, the pursuit of Xe-bearing compounds in the environment of the Earth's mantle has all failed. To explore possible capture and stabilization of Xe by suitable minerals in Earth's mantle will provide critical implications for the origin and evolution of the Earth and even the universe.

In the current manuscript, the authors carried out a comprehensive structure search of stable structures of Xe with common mantle minerals. The results show that FeO₂ and Xe can form thermodynamically stable compounds of Xe₂FeO₂ and XeFe₃O₆ with intriguing chemical bonding pattern in a wide range of pressure-temperature conditions of the Earth's mantle environment that support their viability in vast lower mantle region. In sharp contrast, the other noble gases do not react with major mantle oxides. The results offer new insights into robust materials basis, formation mechanism, and geological viability for elucidating the longstanding "missing xenon paradox". The study has been done professionally, clearly presented, and well written. I recommend its publication in Nature Communications after minor revisions, as suggest below.

1. The authors have provided substantive analyses on phonon dispersions to check for the dynamic stability of these proposed xenon-iron oxides. However, the detailed information for the supercell size and convergence criteria for the calculated dynamical dispersions are missing, which may be useful to the readers.

2. As shown in Fig. 1d, the crystal structure of Xe₂FeO₂ appears to have a pseudo-layered bonding pattern. Although the Van der Waals (vdW) interaction may have little contribution to the energy and transition pressure at high-pressure conditions for these heavy elements, the authors still need to examine the contribution or give proper discussion to the lattice energy arising from vdW interaction.

Reviewer #3 (Remarks to the Author):

Manuscript Review

Robust Xenon-Iron Oxides Predicted in Earth's Lower Mantle
Peng, F. et al.

General Comments

This paper proposes yet another phase for storing the xenon that is "missing" from Earth's atmosphere based on comparisons between solar or primordial noble gas abundances and present-day atmospheric noble gas concentrations. In this case, it is suggested that iron (II) peroxide (FeO₂) will react with elemental xenon in Earth's mantle to produce two possible candidate ternary phases, Xe₂FeO₂ and XeFe₃O₆. These could presumably provide an explanation and mineral repository for Earth's "missing" xenon.

This is a very important question that has been addressed numerous times over the past several decades, and the present work is potentially an interesting contribution to this discussion from a mineral physics perspective. Evolutionary structure-searching methods are used to generate the two candidate compositions and structures, and analysis of the structures and properties of these two candidate phases at P-T conditions appropriate to Earth's mantle by first principles electronic structure and molecular dynamics simulations show that indeed the two phases could be stable at these conditions if formed. The two candidate phases are also adequately shown to be stable with respect to decomposition with respect to a number of different phase combinations. Density and sound velocity calculations are used to compare to currently accepted deep Earth models to show that the presence of these phases in the deep Earth are moderately consistent with observation. The methods are sound and conclusions justified with respect to the stability of the phases at the P-T conditions examined. On balance, the methods used are well-established and not particularly novel at this point in their development. The geochemical/geophysical context is not well developed or supported. The results, however, do potentially constitute an interesting contribution to the literature of high pressure chemistry.

In the opinion of this reviewer, however, work in this area has passed the stage at which simply stating that xenon is present in the Earth and showing by experiment or theoretical modeling that xenon may form stable phases upon combination with existing mineral phases in Earth's interior can provide a useful approach to this problem. At this point, from the geochemical perspective, that the "missing" xenon is present somewhere in the Earth is clear enough. It is now important for authors proposing this type of mineral physics solution to the "missing xenon" problem to establish that in fact free xenon should be present at the depth where, and P-T conditions at which, these existing mantle mineral phases are proposed to react with it. In the case of the present manuscript, this is particularly important, since the presence of FeO₂, whose actual presence in the mantle is highly suspect on geophysical and geochemical grounds anyway, is required for the formation of the candidate phases discussed by the authors. In that respect, it should be clearly established by the authors that free xenon should be available at the specific mantle P-T conditions they propose for reaction with FeO₂ (i.e., the P-T conditions at which goethite presumably transforms to FeO₂). If that information is not available presently in the literature, then the geophysical speculations provided by the authors are out of place and the manuscript should be reformulated as a relevant contribution to xenon chemistry. Indeed, with the first-principles methods employed in this work, there is certainly enough in the way of discussion of redox chemistry with xenon and ferric iron to make a very interesting study.

Specific Comments

1. It seems that over the years, the missing xenon "problem" has become the missing xenon "paradox." Perhaps this makes attempts at solving the problem more pressing or relevant, but this is not a paradox, but rather simply an unsolved problem—perhaps a puzzling problem without a clear answer yet, but still just a scientific problem nonetheless. A paradox exists when the available evidence logically suggests one outcome, when something opposite is actually observed. That is not the case here. The reason why the xenon is missing is understood, we just don't yet know where it is in the Earth.

2. The second to last statement in the abstract, if true, would appear to have solved the missing xenon problem already. In fact, this is clearly not the case, even given the results of the present work. Xe does not react with major mantle oxides either, according to the authors later in the manuscript at line 158, and therefore the fact that other noble gases do not react with major mantle oxides is not a reason for the selective depletion of xenon observed in Earth's atmosphere. This statement should be removed because it is obviously false as well as contradictory.

3. The authors use the Caldwell, et al. (1997) paper to reference work that proposed that Xe is stored in stable compounds within the Earth. In fact, this 1997 paper suggests quite clearly that

the answer to the missing Xe problem lies most likely not in deep mantle or core phases.

4. XeFeO_2 would appear to have the simplest structure and might be predicted over the other two at deep mantle conditions, as minerals tend to become structurally simpler with increasing pressure and temperature. Structurally speaking, why is this composition so unstable compared to the other two? This question should be addressed explicitly.

5. The authors should be more specific about the meaning of “favorable temperature effects” at line 112.

6. Figure 1: Crystal structure representations are inadequate to observe the important details of the structure. What are the coordination geometries around Xe with its coordination numbers of 3 or 6? How does the Xe with CN = 6 in the channel of the XeFe_3O_6 structure result in a soft phonon mode? A better resolution on the phonon dispersion curve would illustrate that soft mode more effectively.

7. The statement at line 163 suggesting that the current results explain the selective depletion of Xe from Earth's atmosphere is not supported by the current work and should be eliminated or heavily qualified. The results provide one plausible explanation for how the missing Xe is sequestered within the Earth, but an explanation for the depletion of Xe from the atmosphere must include a mechanism for how the Xe has been brought from the atmosphere into the mantle to a depth where reaction with FeO_2 is feasible. Likewise, the closing statements at lines 172-178 should be qualified as well.

8. There are numerous errors in usage and grammar throughout the manuscript. It should be carefully proofread by a native English speaker.

In its present form, the manuscript is not suitable for publication in Nature Communications, but may be suitable after major revision and a refocusing of the discussion away from the geochemical perspective if support for the existence of Xe at deep mantle conditions cannot be explicitly documented.

Response to reviewer #1

Reviewer #1 (Remarks to the Author):

This difference between expected and actual abundance of xenon in the Earth's atmosphere is known as the "missing xenon paradox", which has left scientists stumped for decades. Attempt to explain of the paradox, many hypotheses have been proposed, e.g., Xe might be located within the Earth's as stable compounds. However, the pursuit of Xe-bearing compounds in the environment of the Earth's mantle has all failed. To explore possible capture and stabilization of Xe by suitable minerals in Earth's mantle will provide critical implications for the origin and evolution of the Earth and even the universe.

In the current manuscript, the authors carried out a comprehensive structure search of stable structures of Xe with common mantle minerals. The results show that FeO_2 and Xe can form thermodynamically stable compounds of Xe_2FeO_2 and XeFe_3O_6 with intriguing chemical bonding pattern in a wide range of pressure-temperature conditions of the Earth's mantle environment that support their viability in vast lower mantle region. In sharp contrast, the other noble gases do not react with major mantle oxides. The results offer new insights into robust materials basis, formation mechanism, and geological viability for elucidating the longstanding "missing xenon paradox". The study has been done professionally, clearly presented, and well written.

I recommend its publication in Nature Communications after minor revisions, as suggest below:

Reply: We are grateful to the reviewer for the positive assessment of our work, and we appreciate the recommendation for publishing our paper in Nature Communications.

1. The authors have provided substantive analyses on phonon dispersions to check for the dynamic stability of these proposed xenon-iron oxides. However, the detailed information for the supercell size and convergence criteria for the calculated dynamical dispersions are missing, which may be useful to the readers.

Reply: We thank the reviewer for the constructive suggestion. We have now included the following theoretical details starting on line 11 in **Methods** section in the revised manuscript: "To determine the dynamical stability, we performed phonon calculations by the direct supercell method using the Hellmann-Feynman theorem, as implemented

in Phonopy code. The harmonic interatomic force constants are calculated by $3 \times 3 \times 3$ and $3 \times 2 \times 2$ supercells for $P2_1/c$ -Xe₂FeO₂ and $P-1$ -XeFe₃O₆, respectively. Forces were calculated for atomic displacements of 0.01 Å, with a convergence threshold of 1×10^{-5} eV/Å.”

2. As shown in Fig. 1d, the crystal structure of Xe₂FeO₂ appears to have a pseudo-layered bonding pattern. Although the Van der Waals (vdW) interaction may have little contribution to the energy and transition pressure at high-pressure conditions for these heavy elements, the authors still need to examine the contribution or give proper discussion to the lattice energy arising from vdW interaction.

Reply: We thank the reviewer for raising this issue. To examine the contribution of vdW interaction to the lattice energy, we have calculated the enthalpy of formation of Xe₂FeO₂ at high pressures using the vdW-DF2 density functional. Our results show that the enthalpy of formation is less sensitive to the contribution of vdW correction at high-pressure conditions for Xe-FeO₂ compounds, e.g., at 0.6 meV/atom for $P2_1/c$ Xe₂FeO₂ at 135 GPa, thus vdW interaction is not considered in the calculations of lattice energy, as pointed by the reviewer.

We have added the above discussion to line 15 in the **Methods** section in the revised manuscript.

Response to the third reviewer

Reviewer #3 (Remarks to the Author):

General Comments

This paper proposes yet another phase for storing the xenon that is “missing” from Earth’s atmosphere based on comparisons between solar or primordial noble gas abundances and present-day atmospheric noble gas concentrations. In this case, it is suggested that iron (II) peroxide (FeO_2) will react with elemental xenon in Earth’s mantle to produce two possible candidate ternary phases, Xe_2FeO_2 and XeFe_3O_6 . These could presumably provide an explanation and mineral repository for Earth’s “missing” xenon.

This is a very important question that has been addressed numerous times over the past several decades, and the present work is potentially an interesting contribution to this discussion from a mineral physics perspective. Evolutionary structure-searching methods are used to generate the two candidate compositions and structures, and analysis of the structures and properties of these two candidate phases at P-T conditions appropriate to Earth’s mantle by first principles electronic structure and molecular dynamics simulations show that indeed the two phases could be stable at these conditions if formed. The two candidate phases are also adequately shown to be stable with respect to decomposition with respect to a number of different phase combinations. Density and sound velocity calculations are used to compare to currently accepted deep Earth models to show that the presence of these phases in the deep Earth are moderately consistent with observation. The methods are sound and conclusions justified with respect to the stability of the phases at the P-T conditions examined. On balance, the methods used are well-established and not particularly novel at this point in their development. The geochemical/geophysical context is not well developed or supported. The results, however, do potentially constitute an interesting contribution to the literature of high pressure chemistry.

Reply: We appreciate the reviewer’s overall positive assessment on our reported work.

In the opinion of this reviewer, however, work in this area has passed the stage at which simply stating that xenon is present in the Earth and showing by experiment or theoretical modeling that xenon may form stable phases upon combination with

existing mineral phases in Earth's interior can provide a useful approach to this problem. At this point, from the geochemical perspective, that the "missing" xenon is present somewhere in the Earth is clear enough. It is now important for authors proposing this type of mineral physics solution to the "missing xenon" problem to establish that in fact free xenon should be present at the depth where, and P-T conditions at which, these existing mantle mineral phases are proposed to react with it. In the case of the present manuscript, this is particularly important, since the presence of FeO₂, whose actual presence in the mantle is highly suspect on geophysical and geochemical grounds anyway, is required for the formation of the candidate phases discussed by the authors. In that respect, it should be clearly established by the authors that free xenon should be available at the specific mantle P-T conditions they propose for reaction with FeO₂ (i.e., the P-T conditions at which goethite presumably transforms to FeO₂). If that information is not available presently in the literature, then the geophysical speculations provided by the authors are out of place and the manuscript should be reformulated as a relevant contribution to xenon chemistry. Indeed, with the first-principles methods employed in this work, there is certainly enough in the way of discussion of redox chemistry with xenon and ferric iron to make a very interesting study.

Reply: We thank the reviewer for the positive evaluation of our work and constructive suggestions for improving the manuscript. We have carefully considered the reviewer's comments and revised the manuscript accordingly, improving both the physics and the presentation. We have strengthened the description of xenon chemistry at the beginning of the revised manuscript, adding the following discussions.

"The chemical reaction of inert xenon (Xe), a quintessential full-shell element, was earliest predicted by Pauling in 1933 and the first xenon compound was experimentally produced in 1962¹. Then, more xenon compounds were experimentally synthesized at ambient pressure, containing some most electronegative atoms like fluorine²⁻⁵ or oxygen⁶⁻⁹. Subsequently, scientists found that pressure can effectively improve the chemical reactivity of Xe¹⁰⁻¹⁷. At moderate pressures, solid xenon forms weakly bonded compounds with other species, e.g., with H₂O¹⁰ and O₂^{11,12} at 1 and 3 GPa, respectively. Strikingly, several novel Xe compounds with unusual stoichiometries have been found to be thermodynamically stable at high pressures, e.g., Xe oxides^{13,14}, Xe nitrides¹⁵, xenon-hydrogen¹⁶, and Xe-Mg compounds¹⁷."

1. Bartlett, N. Xenon Hexafluoroplatinate(v) Xe⁺[PtF₆]. *Proc. Chem. Soc.* **1**, 218 (1962).
2. Chernick, C. L. et al. Fluorine Compounds of Xenon and Radon. *Science* **138**, 136 (1962).

3. Claassen, H. H., Selig, H. & Malm, J. G. Xenon Tetrafluoride. *J. Am. Chem. Soc.* **84**, 3593 (1962).
4. Gavin JR, R. M. & Bartell, L. S. Molecular Structure of XeF₆. I. Analysis of Electron-Diffraction Intensities, *J. Chem. Phys.* **48**, 2460 (1968).
5. Hoppe, R., Daehne, W., Mattauch, H. & Roedder, K. Fluorination of Xenon. *Angew. Chem. Int. Ed. Engl.* **1**, 599 (1962).
6. Smith, D. F. Xenon Trioxide. *J. Am. Chem. Soc.* **85**, 816 (1963).
7. Templeton, D. H., Zalkin, A., Forrester, J. D. & Williamson, S. M. Crystal and Molecular Structure of Xenon Trioxide. *J. Am. Chem. Soc.* **85**, 817 (1963).
8. Huston, J. L., Studier, M. H. & Sloth, E. N. Xenon Tetroxide: Mass Spectrum. *Science* **143**, 1161 (1964).
9. Selig, H., Claassen, H. H., Chernick, C. L., Malm, J. G. & Huston, J. L. Xenon Tetroxide: Preparation and Some Properties. *Science* **143**, 1322 (1964).
10. Sanloup, C., Mao, H.-K. & Hemley, R. J. High-pressure transformations in xenon hydrates. *Proc. Natl. Acad. Sci. USA* **99**, 25 (2002).
11. Dewaele, A., Loubeyre, P., Dumas, P. & Mezouar, M. Oxygen impurities reduce the metallization pressure of xenon. *Phys. Rev. B* **86**, 014103 (2012).
12. Weck, G., Dewaele, A. & Loubeyre, P. Oxygen/noble gas binary phase diagrams at 296 K and high pressures. *Phys. Rev. B* **82**, 014112 (2010).
13. Zhu, Q., Jung, D. Y., Oganov, A. R., Glass, C. W., Gattiand, C. & Lyakhov, A. O. Stability of Xenon Oxides at High Pressures. *Nat. Chem.* **5**, 61 (2012).
14. Hermann, A. & Schwerdtfeger, P. Xenon Suboxides Stable under Pressure. *J. Phys. Chem. Lett.* **5**, 4336 (2014).
15. Peng, F., Wang, Y., Wang, H., Zhang, Y. & Ma, Y. Stable Xenon Nitride at High Pressures. *Phys. Rev. B* **92**, 094104 (2015).
16. Somayazulu, M. et al. Pressure-induced bonding and compound formation in xenon–hydrogen solids. *Nat. Chem.* **2**, 50 (2010).
17. Miao, M. S. et al. Anionic Chemistry of Noble Gases: Formation of Mg–NG (NG = Xe, Kr, Ar) Compounds under Pressure. *J. Am. Chem. Soc.* **137**, 14122 (2015).

Below we clarify the technical issues raised by the reviewer.

Specific Comments:

1. It seems that over the years, the missing xenon “problem” has become the missing xenon “paradox.” Perhaps this makes attempts at solving the problem more pressing or relevant, but this is not a paradox, but rather simply an unsolved problem —perhaps a puzzling problem without a clear answer yet, but still just a scientific problem nonetheless. A paradox exists when the available evidence logically suggests one outcome, when something opposite is actually observed. That is not the case here. The reason why the xenon is missing is understood, we just don’t yet know where it is in the Earth.

Reply: We thank the reviewer for clarifying the issue. We have changed the “missing xenon paradox” into the “missing xenon problem” in our revised manuscript.

2. The second to last statement in the abstract, if true, would appear to have solved the missing xenon problem already. In fact, this is clearly not the case, even given the results of the present work. Xe does not react with major mantle oxides either, according to the authors later in the manuscript at line 158, and therefore the fact that other noble gases do not react with major mantle oxides is not a reason for the selective depletion of xenon observed in Earth’s atmosphere. This statement should be removed because it is obviously false as well as contradictory.

Reply: We are thankful for the reviewer’s constructive suggestion. In the revised abstract, we have appropriately rewritten the sentence of

“Meanwhile, major mantle oxides do not react with Kr, Ar, and Ne, explaining the selective depletion of Xe in Earth’s atmosphere”

to

“Meanwhile, Fe oxides do not react with Kr, Ar, and Ne, suggesting the distinct possibility that Xe could be captured in Earth’s lower mantle.”

3. The authors use the Caldwell, et al. (1997) paper to reference work that proposed that Xe is stored in stable compounds within the Earth. In fact, this 1997 paper suggests quite clearly that the answer to the missing Xe problem lies most likely not in deep mantle or core phases.

Reply: We thank the reviewer for a careful reading and correction. We have made the adjustment and cited this reference as Ref. [22] in line 5, paragraph 2, in the section of **Introduction**.

4. XeFeO₂ would appear to have the simplest structure and might be predicted over the other two at deep mantle conditions, as minerals tend to become structurally simpler with increasing pressure and temperature. Structurally speaking, why is this composition so unstable compared to the other two? This question should be addressed explicitly.

Reply: We thank the reviewer for making this suggestion. In response, we now present in **Supplementary Figure 1** the relative enthalpies ΔH , internal energies ΔU , and pressure-volume terms $\Delta(PV)$ to examine the mechanism of thermodynamic stability for these FeO₂-Xe compounds. Our results show that the pressure-volume terms, associated with packing efficiency, make similar contributions to lowering the enthalpy for XeFe₃O₆ and XeFeO₂, while the unfavorable internal energy, associated with bonding enhancement, leads to a positive formation enthalpy for XeFeO₂.

Although the value of pressure-volume term of Xe_2FeO_2 is much lower than those of XeFe_3O_6 and XeFeO_2 , it still offsets the relatively weak negative effect of internal energy to enthalpy, thus yielding its thermodynamical stability with formation enthalpy lying on the convex hull. To address this question, we added a figure [Supplementary Figure 1] in the Supplementary Information and give the above discussion in the figure caption.

[New Supplementary Figure 1 in revised Supplementary Information]. **Calculated energetic terms of FeO_2 -Xe compounds and the XeFeO_2 structure at 150 GPa.** (a) The enthalpies ΔH , the internal energies ΔU , and the pressure-volume term $\Delta(PV)$ for the Xe-Fe oxides. (b) The polyhedral views of theoretically predicted XeFeO_2 structure. We discuss the energetic terms to examine the mechanism of thermodynamic stability for these FeO_2 -Xe compounds. The pressure-volume terms, associated with packing efficiency, make similar contributions to lowering the enthalpy for XeFe_3O_6 and XeFeO_2 , while the unfavorable internal energy, associated with bonding enhancement, leads to a positive formation enthalpy for XeFeO_2 . Although the value of pressure-volume term of Xe_2FeO_2 is much lower than those of XeFe_3O_6 and XeFeO_2 , it still offsets the relatively weak negative effect of internal energy to enthalpy, thus yielding its thermodynamic stability with formation enthalpy lying on the convex hull.

5. The authors should be more specific about the meaning of “favorable temperature effects” at line 112.

Reply: To avoid ambiguity, we have changed the sentence

“Favorable temperature effects reduce Gibbs free energy of the Xe-Fe oxides even more relative to their separate constituents, ...”

into

“The temperature effect (i.e., thermal vibration of atomic positions) are favorable to reduce Gibbs free energy of the Xe-Fe oxides even more relative to their separate constituents,”

in our revised manuscript.

6. Figure 1: Crystal structure representations are inadequate to observe the important details of the structure. What are the coordination geometries around Xe with its coordination numbers of 3 or 6? How does the Xe with CN = 6 in the channel of the XeFe_3O_6 structure result in a soft phonon mode? A better resolution on the phonon dispersion curve would illustrate that soft mode more effectively.

Reply: We thank the reviewer for making this constructive suggestion. To present clearly the coordination numbers of Xe, the units of Xe-O for Xe_2FeO_2 and XeFe_3O_6 are shown in Fig. 1(d) and 1(e) in the revised manuscript. The coordination geometries around Xe with the coordination numbers of Xe_2FeO_2 and XeFe_3O_6 are 3 and 6, respectively.

Indeed, the vibrational modes close to zero for XeFe_3O_6 stem from the relatively large internal volume for Xe atom that leads to the concomitant longer bond lengths and weak interaction between Xe atoms and O atoms. The maximum distance between Xe and O atom is 2.52 Å in XeFe_3O_6 at 150 GPa, which is much longer than that in Xe oxides (e.g. ~2.37 Å in Xe_2O_5) and Xe_2FeO_2 with CN=3 of Xe (2.42 Å) at the same pressure. As a result, Xe-O bonds of Xe_2FeO_2 are much stronger than that of XeFe_3O_6 . To explore possible dynamical instabilities, we have moved one Xe atom along the channels of the Fe-O framework (the vibration eigenvector of the soft phonon mode) while fixing all other atoms. The calculated static energy presented in **Supplementary Figure 4** shows that the original structure is located at an energy minimum on the energy surface, signifying its dynamical stability. The curve is smooth near the equilibrium position (e.g., < 0.2 Å), making it easier for the Xe atoms to move about along the eigenvector of the low-frequency phonon mode, in agreement with the theoretical phonon spectrum. Meanwhile, the curve has a rather high energy cost at large displacements and the phase transition is impeded by a

kinetic barrier of ~ 26.8 meV/atom at 150 GPa and zero temperature, indicating that it is difficult for the Xe atoms to escape from the lattice. With the consideration of temperature effect, our AIMD calculations also reveal that Xe atoms can only vibrate near the balanced positions below melting points. To address this question, we have appropriately modified **Supplementary Figure 4** in the Supplementary Information and give the above discussion in the figure caption.

[New Supplementary Figure 4 in revised Supplementary Information]. **The vibrational analyses for XeFe_3O_6 .** (a) The internal energy difference at various Xe positions along the channels (the vibration eigenvector of the soft phonon mode) of the Fe-O framework at 150 GPa and zero temperature. The original structure is located at an energy minimum on the energy surface, signifying its dynamical stability. (b) The internal energy difference of XeFe_3O_6 at various Xe positions along the channels near the equilibrium position. (c) The theoretical phonon spectrum at the low-frequency region. (d) The snapshot for the Fe-O framework with Xe atom in the channel. The vibrational modes close to zero for XeFe_3O_6 stem from the relatively large internal volume for Xe atom that leads to the concomitant longer bond lengths and weak interaction between Xe atoms and O atoms. The maximum distance between Xe and O atom is 2.52 Å in XeFe_3O_6 at 150 GPa, which is much longer than that of Xe oxides (e.g. ~ 2.37 Å in Xe_2O_5) and Xe_2FeO_2 with CN=3 of Xe (2.42 Å) at the same pressure. As a result, Xe-O bonds of Xe_2FeO_2 are much stronger than that of XeFe_3O_6 . To explore the possible dynamical instabilities, we have move one Xe atom along the channels of the Fe-O framework (the vibration eigenvector of the soft phonon mode) while fixing the other atoms. The calculated static energy in **Supplementary Figure 4** shows that the original structure is located at an energy minimum on the energy surface, signifying its dynamical stability. The curve is

smooth near the equilibrium position (e.g., $< 0.2 \text{ \AA}$), making it easier for the Xe atoms to move about along the eigenvector of the low-frequency phonon mode, in agreement with the theoretical phonon spectrum. Meanwhile, the curve has a rather high energy cost at large displacements, and the phase transition is impeded by kinetic barrier of $\sim 26.8 \text{ meV/atom}$ at 150 GPa and zero temperature, indicating that it is difficult for the Xe atoms to escape from the lattice. With the consideration of temperature effect, our AIMD calculations also reveal that Xe atoms can only vibrate near the balanced positions below melting points.

7. The statement at line 163 suggesting that the current results explain the selective depletion of Xe from Earth's atmosphere is not supported by the current work and should be eliminated or heavily qualified. The results provide one plausible explanation for how the missing Xe is sequestered within the Earth, but an explanation for the depletion of Xe from the atmosphere must include a mechanism for how the Xe has been brought from the atmosphere into the mantle to a depth where reaction with FeO_2 is feasible. Likewise, the closing statements at lines 172-178 should be qualified as well.

Reply: We thank the reviewer for this constructive suggestion and agree with the reviewers' viewpoints. We have changed the statements in line 163 in our previous version of manuscript:

“This result explains the selective depletion of Xe from Earth's atmosphere.”.

to

“These results provide the possibility that Xe could be the sole inert element for reacting with deep-Earth constituents under mantle conditions.”

We have also deleted a sentence to weaken the statement in line 175-176 in our previous version of manuscript:

“These findings offer new insights into key material composition and physics mechanism for elucidating the longstanding “missing xenon paradox”.”

8. There are numerous errors in usage and grammar throughout the manuscript. It should be carefully proofread by a native English speaker.

Reply: In response to the reviewer's suggestion, we have polished our manuscript carefully.

In its present form, the manuscript is not suitable for publication in Nature Communications, but may be suitable after major revision and a refocusing of the discussion away from the geochemical perspective if support for the existence of Xe at deep mantle conditions cannot be explicitly documented.

Reply: We thank the reviewer for a careful reading of our manuscript and for making numerous constructive suggestions for its improvement. Following these suggestions, we have made extensive revisions accordingly, and we hope that these revisions are satisfactory and that the revised manuscript is now acceptable for publication in Nature Communications.

REVIEWERS' COMMENTS

Reviewer #1 (Remarks to the Author):

This revision has improved the quality substantially, addressing the most concerns raised by the referees. Clearly, the paper presents interesting new results that may have some implications for the experimental studies of Xenon-Iron Oxides. Thus, I would recommend the manuscript for publication based on the results providing a valuable understanding of the phase diagram in Xenon-Iron Oxides.

Reviewer #3 (Remarks to the Author):

Overall, the points raised in the original review have been adequately addressed, with the exception of specific point #7:

As stated in the initial review, if the proposed compounds are to be invoked as a potential solution to the "missing Xe problem" then the presence of Xe in the lower mantle must be established from the existing geochemical literature and clearly stated, if it in fact exists. Otherwise the authors have not done their homework and the paper is not appropriate as a contribution to the solution of the missing Xe problem. If this evidence from the geochemical literature does not exist, then this must be clearly stated in the text. In this paper, no evidence from the literature has been provided for Xe existing in the lower mantle from 24 GPa / 1900 K - 145 GPa / 3500 K. Therefore, the most that can be said here is that *** IF *** Xe exists in the lower mantle at the same pressures as FeO₂, then the potential exists for these compounds to form, and these could provide the repository for the atmosphere's missing Xe. This should be explicitly stated in the abstract and in the text. It should be further pointed out that the P-T stability range of FeO₂ has not been established, and so it should be clearly stated that this is the case. Otherwise, the authors are taking what is actually a very thorough contribution to xenon chemistry and evading the real issue that is necessary for it to be considered as a solution to the missing Xe problem. The authors should not be permitted to discuss their results as though these two key requirements for their proposed compounds are actually confirmed. This would be a matter for the editor to decide.

The paper is suitable for publication after this minor revision.

Given the preceding comments, I would also suggest that the title be changed to something like "Xenon Iron Oxides Predicted as Potential Xe Hosts in Earth's Lower Mantle"

Response to reviewer #1

Reviewer #1 (Remarks to the Author):

This revision has improved the quality substantially, addressing the most concerns raised by the referees. Clearly, the paper presents interesting new results that may have some implications for the experimental studies of Xenon-Iron Oxides. Thus, I would recommend the manuscript for publication based on the results providing a valuable understanding of the phase diagram in Xenon-Iron Oxides.

Reply: We appreciate the reviewer's positive assessments of our reported work and the recommendation for publishing our paper in *Nature Communications*.

Response to reviewer #3

Reviewer #3 (Remarks to the Author):

Overall, the points raised in the original review have been adequately addressed, with the exception of specific point #7:

*As stated in the initial review, if the proposed compounds are to be invoked as a potential solution to the "missing Xe problem" then the presence of Xe in the lower mantle must be established from the existing geochemical literature and clearly stated, if it in fact exists. Otherwise the authors have not done their homework and the paper is not appropriate as a contribution to the solution of the missing Xe problem. If this evidence from the geochemical literature does not exist, then this must be clearly stated in the text. In this paper, no evidence from the literature has been provided for Xe existing in the lower mantle from 24 GPa / 1900 K - 145 GPa / 3500 K. Therefore, the most that can be said here is that *** IF *** Xe exists in the lower mantle at the same pressures as FeO₂, then the potential exists for these compounds to form, and these could provide the repository for the atmosphere's missing Xe. This should be explicitly stated in the abstract and in the text. It should be further pointed out that the P-T stability range of FeO₂ has not been established, and so it should be clearly stated that this is the case. Otherwise, the authors are taking what is actually a very thorough contribution to xenon chemistry and evading the real issue that is necessary for it to be considered as a solution to the missing Xe problem. The authors should not be permitted to discuss their results as though these two key requirements for their*

proposed compounds are actually confirmed. This would be a matter for the editor to decide.

Reply: We appreciate the reviewer's professional considerations and constructive suggestions for improving our manuscripts. We have carefully considered the reviewer's comments and revised the manuscript accordingly. In the revised manuscript, we have added the following sentence

"It means that if Xe exists in the lower mantle at the same pressures as FeO₂, xenon iron oxides are predicted as a potential Xe hosts in Earth's lower mantle and could provide the repository for the atmosphere's missing Xe."

at line 9 in the section of **Abstract**.

We have also rewritten the sentence

"Recently discovered FeO₂ synthesized at lower mantle conditions³⁵ offers an intriguing new possibility."

to

"Recently discovered FeO₂ synthesized at lower mantle conditions³⁵ and stabilized above 74 GPa in theoretical calculation³⁶, offer an intriguing new possibility."

at line 2, paragraph 3 in **Introduction**

and supplemented the sentence

"(actually P-T stability range of FeO₂ has still not been completely established in experiments)"

at line 5, paragraph 3, in the section of **Introduction**.

The paper is suitable for publication after this minor revision.

Reply: We are grateful to the reviewer's recommendation for publishing our paper in *Nature Communications* after this minor revision.

Given the preceding comments, I would also suggest that the title be changed to something like "Xenon Iron Oxides Predicted as Potential Xe Hosts in Earth's Lower Mantle".

Reply: We thank the reviewer for the constructive suggestion. In response, we have changed the title to "Xenon Iron Oxides Predicted as Potential Xe Hosts in Earth's Lower Mantle".